# Assessing the Competence of Nursing Students in Clinical Practice: The Clinical Preceptors’ Perspective

**DOI:** 10.3390/healthcare12101031

**Published:** 2024-05-16

**Authors:** Watin Arif Alkhelaiwi, Marian Traynor, Katherine Rogers, Iseult Wilson

**Affiliations:** 1Nursing Department, Faculty of Applied Medical Science, Jouf University, Sakaka 72388, Aljouf Region, Saudi Arabia; 2School of Nursing and Midwifery, Queen’s University Belfast, Belfast BT9 7BL, UK; m.traynor@qub.ac.uk (M.T.); k.rogers@qub.ac.uk (K.R.); i.wilson@qub.ac.uk (I.W.); 3College of Nursing and Midwifery, Mohammed Bin Rashid University, Dubai P.O. Box 505055, United Arab Emirates

**Keywords:** competence, clinical competence, competency assessment, clinical assessment tools, preceptor, mentor, nursing, nurses

## Abstract

Nursing students’ integration of theoretical knowledge and practical abilities is facilitated by their practice of nursing skills in a clinical environment. A key role of preceptors is to assess the learning goals that nursing students must meet while participating in clinical practice. Consequently, the purpose of this study was to explore the current evidence in relation to competency assessment and assessment approaches, and the willingness of preceptors for assessing nursing students’ competency in a clinical setting. The scoping review used the five-stage methodological framework that was developed by Arksey and O’Malley, as well as the Preferred Reporting Items for Systematic Reviews and Meta-Analyses Extension for Scoping Reviews. Relevant studies were searched by applying a comprehensive literature search strategy up to April 2024 across the following databases: CINAHL, OVID MEDLINE, EMBASE, and PUBMED. A total of 11,297 studies published between 2000 and April 2024 were revealed, and 38 were eligible for inclusion, which the research team categorised into three main themes: definitions of competence, tools for assessing competence and preceptors’ and mentors’ viewpoints in relation to the assessment of nursing students’ competence. This review established that there are a multitude of quantitative instruments available to assess clinical competence; however, a lack of consistency among assessment instruments and approaches between countries and higher education institutions is prevalent. Existing research evidence suggests that the preceptors carried out the assessment process clinically and they found difficulties in documenting assessment. The assessing of nursing students’ competency and the complexity of assessment is a concern for educators and mentors worldwide. The main concern centers around issues such as the interpretation of competence and complex measurement tools.

## 1. Introduction

Clinical assessment is an important issue in clinical nursing education programmes. Assessment provides an opportunity for nursing students to acquire the required knowledge through practice and assessment in a variety of clinical settings [1,2]. Clinical practice assists nursing students in improving nursing skills and adapting them to a variety of professional roles and clinical places [2]. Further, it contributes to the integration of theoretical knowledge and clinical skills for nursing students [2].

Most nursing scholars Expressed a broad view of competence as a virtue, as well as knowledge. Competence contains broad features that pertain to the capacity to carry out a task under various conditions and provide desired results [3,4]. It is further accomplished through good abilities, skills, attitudes, and values in the same line with ethical behaviour and the effective delivery of quality services [5,6,7,8]. Nursing competence is a crucial capability required to realise the responsibilities of nursing. Therefore, it is important to clearly define the concept of nursing competence and establish a fundamental definition for nursing education curricula. It would be helpful to understand how nurses develop competence in order to enhance continuous professional development and to support improvements in nursing quality [9].

Efficient mentoring in clinical environments assists nursing students in promoting the required competence and improving the integration of theory and practice [10]. The mentoring of nursing students is performed by nursing staff known as ‘nursing preceptors’ in Saudi Arabia (SA) [11], Ireland [12], and in the United States and Sweden [13]. They are known in the United Kingdom as ‘nursing mentors’ [14], and as ‘buddy nurses’ in Australia [15].

Nursing preceptors have a significant role in clinical nursing education. The role of nursing preceptors in clinical nursing education includes supervising nursing students who are enrolled in a rigorous clinical practicum program [16]. Among the responsibilities of nursing preceptors is guiding nursing students through the integration of theoretical knowledge into clinical practice, guiding and teaching practical nursing skills, and improving skills of problem-solving and critical thinking [17]. Preceptors are further included in assessing nursing students’ competence [18]. They have an essential responsibility in assessing the learning outcomes that must be fulfilled by nursing students in clinical environments [19].

Considering the importance of the preceptors’ roles, it might be helpful to explore how nursing preceptors view the assessment of competence. This would increase perception of the significance of competence assessment strategies in the nursing field and thus contribute to making the outcomes of nursing program more eligible and qualified.

It is essential to assess the competence of nursing students with professional practitioners in order to examine whether nursing students have developed sufficient competence levels for providing safe nursing care to patients [20,21,22]. Performance-based systems can be used to assess competency by using a variety of tools and allowing students to achieve certain levels of competence [21,22,23]. Using a valid assessment competence tool may be helpful in promoting and developing good-quality nursing education [24]. 

The complexity of assessing competence is still a concern for nursing educators and preceptors [22]. There is inconsistency in terms of assessment approaches and instruments [17] and a lack of validity and reliability of assessment instruments for measuring competence in clinical practice [25]. Part of the nurse preceptor’s role is to assess the competence of nursing students in order to make them ready for professional responsibilities and future tasks. However, preceptors find it difficult to assess students’ competencies in an objective manner [2,18,26]. Moreover, they encounter several challenges during the assessment of nursing students’ competencies, including responsibilities associated with conflict, work stress, overload work, and ambiguity of assessment documents [16,27].

The aim of the scoping review is to explore the current evidence in relation to competency assessment and assessment approaches, and the willingness of preceptors for assessing nursing students’ competency in clinical setting. 

## 2. Materials and Methods

A scoping review is a response to a particular methodology that involves identifying and evaluating previous research on a certain subject based on predetermined eligibility standards. The purpose of the scoping review is to summarise, analyse, and report the findings that clearly address a particular research question. A preliminary search of the International Databases to registered Systematic Reviews such as PROSPERO, OSF, and INPLASY^®^ did not identify any reviews on the topic. Preferred Reporting Items for Systematic Reviews and Meta-Analyses (PRISMA) 2020 guidelines were adopted for this study due to their applicability, as they are primarily intended to assess systematic reviews of studies, irrespective of the included studies’ designs [28,29]. Specifically, this scoping review was conducted using the PRISMA extension for Scoping Reviews (PRISMA-ScR) checklist [29]; the scoping review’s protocol has been registered at Open Science Framework (OSF). 

The methodology of the scoping review included the five-stage methodological framework that is designed by Arksey and O’Malley (2005) [30] and then advanced by Levac et al. (2010) [28]. These five stages are: (1) identifying the research question, (2) identifying relevant studies, (3) study selection, (4) charting the data, and (5) collating, summarising, and reporting the results [28,31]. The guidelines of the Preferred Reporting Items for Systematic Reviews and Meta-Analyses extension for Scoping Reviews (PRISMA-ScR) checklist was followed to ensure rigour in reporting the results of the study [29]. 

### 2.1. Identifying the Research Questions 

To achieve the aim of this scoping review, the following research questions were formulated after using the PCC framework: 1. What are the definitions of competence? 2. What are tools used for assessing competence? 3. What are the viewpoints of nursing preceptors and mentors regarding the assessment of competence? 

### 2.2. Identifying Relevant Studies

After the research questions were created, in order to optimise the evidence recovered from the databases, the search strategy of the literature was based on the PCC frame-work, involving the dividing of the research question in accordance with the components that follow: Population (P), Concepts (C) and Context (C). Table 1 presents the PCC model:

An analysis of the research questions’ terms was carried out to help guide the scoping review’s search strategy according to the PCC framework. A comprehensive list of terms was created for the research strategy in consultation with a research team and a nursing librarian after identifying them according to the terms MeSH (Medical Subject Headings) and DeCS (Health sciences desCriptors). The search strategy was conducted using the terms and the relevant synonyms were identified as follows in Table 2.

Relevant studies were searched by applying a comprehensive literature search strategy up to April 2024 across the following four databases: CINAHL, OVID MEDLINE, EMBASE, and PUBMED. Time limitations during the study searching were set for the last 5 years, then extended to 10, 15 and finally 20 years, however, there were no studies older than 15 years that met the inclusion criteria. Further, restricting the search date is a legitimate and fast way to provide reliable information for prompt decision-making. Xu et al. (2021) [32] conducted a study to investigate the accuracy and workload of search date limitations, and the search results and the results identified fell within a maximum tolerance of 5% and 10% for magnitude bias, indicating that a limitation on the last 20 years can save the most effort while still achieving high accuracy. The databases were selected with the goal of locating studies pertinent to the topic of the scoping literature review, and the reasons for choosing these databases was because of their widespread interdisciplinary coverage and worldwide renown. Furthermore, a manual search of the reference lists of eligible studies was also used. Studies containing descriptors linked to the terms MeSH (Medical Subject Headings) and DeCS (Health sciences desCriptors) were chosen based on the eligibility criteria. The MeSH and DeCS descriptors were used to identify the terms Preceptors, Competency, and Nursing. These terms and their synonyms were then combined with a Boolean operator to create a database search that would accomplish the proposed objectives. The searches with descriptors and Boolean AND and OR operators are presented in Table 3.

The scoping review was conducted through many stages including organising, summarising, and integrating the findings from various studies to develop a coherent understanding of the competency assessment in clinical practice. The research team commenced by identifying common themes, patterns, and gaps in the included studies. Then, they created a table, and highlighted the key findings to present the information effectively. Finally, they ensured the synthesis was clear, concise, and accurately represented the breadth of the literature that were reviewed.

### 2.3. Study Selection

Studies were selected according to inclusion and exclusion criteria. Inclusion criteria were as follows: (1) All studies concerning the nursing assessment of competence and competence. (2) Studies examining nursing preceptors’ experiences in the nursing assessment of competence and their perspectives of making decisions in relation to the assessment of nursing students’ competence. (3) Studies which identified present instruments or had a developed instrument for assessing nursing students’ competence. (4) Peer-reviewed studies that have full text, open access studies and those published in academic journals and in the English language from 2000–2024, (5) Peer-reviewed reviews including systematic, scoping, narrative, and integrative reviews. The exclusion criteria in this scoping review were as follows: (1) Studies concerning assessment of healthcare students’ competence. (2) Studies concerning the perspectives of nursing educators and nursing students in relation to assessment of competence. (3) Conference abstracts, posters, and opinion studies. The benefit of a scoping review is that it offers the opportunity to include all types of studies, and so these are included in the interest of thoroughness.

### 2.4. Charting the Data

The studies that met the inclusion criteria were uploaded into the reference manager Mendeley, and this application was used to remove duplicated studies. Included studies were downloaded into the Mendeley application to remove duplication. The process of data analysis started with the first author using a data extraction form and inputting the data into an Excel spreadsheet. The following categories were created to extract the data for each of the included studies, when eligible: Author(s), year, aim, country of the study, key words, study design, participants (sample or number of studies), instruments, and key findings that relate to the scoping review objectives, and these were then reviewed and verified by the research team.

### 2.5. Collating, Summarising, and Reporting the Results

There were 11,297 studies found in the initial search results across all databases. One author initially reviewed the titles and abstracts; 191 studies were included while 8043 were excluded. One author conducted a second screening of these 191 studies’ titles and abstracts, and 58 studies were included. The 58 studies’ titles and abstracts were also reviewed by the research team and 20 studies did not meet the inclusion criteria, so were excluded. The first author retrieved the full texts of the included 38 studies and the research team reviewed the data extraction tables independently. The research team categorised the 38 studies into 3 main themes. The results of the literature search and study screening process are presented in the PRISMA-ScR flow diagram [29] in Figure 1. 

## 3. Results

### 3.1. Overview of Included Studies 

#### Description of Studies

This scoping review included thirty-eight research studies published between 2008 and 2023 (Table 4). These studies took place in 19 countries: Ireland (n = 8), Australia (n = 5), UK (n = 3), with two studies in each of the following countries: Finland, Singapore, Korea, Taiwan, and USA and one study in each of the following countries: Sweden, Spain, Slovenia, Turkey, Japan, China, Thailand, Jorden, Iran, and New Zealand. Two studies were conducted in both Australia and Canada (Table 5). These studies used multiple research designs; a total of 15 were quantitative research studies, 6 were qualitative research studies, 5 were mixed methods research studies, and 12 were literature reviews. Nearly half of the studies (n = 17) used questionnaires as data collection methods, whereas 9 studies employed individual or focus group interviews as methods for data collection. The participants were diverse, including preceptors, nurses, mentors, clinical assessors, clinical educators, nursing teachers, practice educators, and nursing students.

The themes category as follows: definitions of competence, tools for assessing competence, and preceptors’ and mentors’ viewpoints in relation to the assessment of nursing students’ competence. Following data extraction, the data were then mapped against the three stated research questions. For example, question one sought to define competence. Four studies [9,34,35,36] contributed to this, and their findings are presented below. Question two sought to identify the tools which are used for assessing competence. Eighteen studies [22,24,37,38,39,40,41,42,43,44,45,46,47,48,49,50,51,52] contributed to find existing tools that were used to assess competency, as well as specific assessment competence tools developed for this purpose. Question three sought to examine preceptors’ and mentors’ viewpoints related to the assessment of nursing students’ competence. Sixteen studies [2,12,18,21,25,27,53,54,55,56,57,58,59,60,61,62] contributed to examining this. 

### 3.2. The Definitions of Competence 

In this scoping review, four of the included studies provided definitions of competence [9,34,35,36] and these varied based on the extensive literature review, Nehrir et al. (2016) [34] stated that nursing students’ competency is ‘‘the individual experiences, dynamic process, and positive interactive social and beneficial changes in the equality of one’s professional life that cause meta-cognitive abilities, touch reality, motivation, decision making, job involvement, professional authority, self-confidence, knowledge and professional skills”. Mrayyan et al. (2023) [36] stated that in the previous literature, competency in nursing practice was defined by knowledge, self-evaluation, and dynamic state. In addition, based on a Bachelor of Nursing degree course [35], competency was analysed as ‘‘the existence of a hierarchy of competencies that prioritises soft skills over intellectual and technical skills; the appearance of skills as personal qualities or individual attributes; and the absence of context in assessment”. Fukada (2018) [9] argues that the concept of nursing competency has not been fully developed. Thus, challenges remain in establishing an agreed definition and structure of nursing competency. Whilst there is no single agreed definition of competence, there is the agreement that competence includes a range of complex attributes such as theoretical and intellectual skills (e.g., knowledge and critical thinking), practical and behavioural skills (e.g., the ability to perform a skill safely and effectively), and personal and professional attributes (e.g., ethical practices and values).

### 3.3. Tools Are Used for Assessing Competence 

The studies that explored tools used for assessing competence were analysed in two main subcategories including the identification of existing clinical nursing assessment competency tools, and the evaluation of specific tools that are developed for measuring clinical nursing assessment competency. There were diverse modes of assessing nursing competence assessment. Six studies identified the existing clinical nursing assessment competency tools [22,24,37,38,39,40] (Table 4 (2) theme 2). The last 12 studies in this category developed and evaluated specific nursing assessment competency instruments, such as the Nursing Students Competence Instrument (NSCI) [44], Advanced Practice Nursing Competency Assessment Instrument (APNCAI) [43], Amalgamated Student Assessment in Practice (ASAP) model and tool [46], Shared Specialist Placement Document (SSPD) tool [45], Australian Nursing Standards Assessment Tool (ANSAT) instrument [47], Competency Assessment Instrument [51], Nursing Practice Readiness Scale [49], Health Education Competency Scale (HECS) [48], Nurse Professional Competence (NPC) Scale [50] and the Nurse Competence Scale [52]. Whilst there are multiple different assessment competence tools across different countries, and assessment competence tools are developed based on the national standards, certifications, and licence criteria in each country, there is agreement that the tools focus on many domains, such as professional attributes, ethical practices, communication and interpersonal relationships, nursing processes, and critical thinking and reasoning. Also, using a valid assessment competence tool to assess certain levels of competence in students. This would be helpful in promoting and developing good-quality nursing education and achieving certain levels of competence.

### 3.4. The Viewpoints of Nursing Preceptors and Mentors of Assessment of Competence

This theme discussed the preceptors’ and mentors’ viewpoints of the assessment of competence in relation to preceptors’ experiences of the competency assessment process, the preceptors’ challenges of the competency assessment process, and mentor judgements and preceptors’ decision-making process/failing. Ten studies explored nursing preceptors’ perspectives of assessment competency in various research methods [2,18,21,25,53,54,55,56,57,62]. The preceptors indicated that they found difficulties in understanding the language used in the document for assessing competence. The wording lacked clarity and needed to be defined. Also, the need for a valid and reliable clinical assessment tool was required from preceptors. Two studies explored the challenges that nursing preceptors face during assessing the nursing competence [21,27]. These two studies’ findings reported that there are challenges face preceptors during the assessment process, such as difficulties in the language used for describing competencies; distinguishing among competency levels; the lack of constructive and clear feedback to nursing students; and the lack of transparent and explicit criteria, hinders the accurate and fair assessment of students. Further, numerous preceptors were inexperienced, did not completely understand the assessment procedure, and did not use all of the required assessment techniques while assessing students in clinical practice. Four studies considered preceptors’ decision-making process and mentor judgements for assessing nursing students’ competence [58,59,60,61]. Whilst the findings of the preceptors’ experiences about the assessment competence are indicated from preceptors from different countries, in various studies, and several clinical environments, there is agreement that the role of preceptors is complicated and they face challenges during the preceptorship process. 

## 4. Discussion

The scoping review aimed to explore the existing evidence regarding the assessment of competence and assessment methods, and the preparedness of nursing preceptors to assess the competency of nursing students in clinical practice. The scoping review, comprising 38 studies, examined definitions of competence, assessment tools in current use, and the viewpoints of nursing preceptors and mentors of assessment of competence. 

Variations in competence definitions were evident, reflecting contextual differences in practice [34]. Despite numerous proposed definitions, clarity remains elusive, necessitating a simple and coherent definition adaptable across institutions. Consensus suggests competence comprises multifaceted qualities, including knowledge, skills, and professional attributes, in order to improve the assessment of nursing competence for preceptors and nursing students in the clinical practice in a clear method. Moreover, identifying nursing competence promotes the continuous development of professional nursing and nursing quality [9].

The competence definitions were derived from a variety of resources and instruments. These variations in the definitions of competence result in a variety of resources to describe competency, which were defined locally with a variety of competency categories. Furthermore, it has been observed that there has been a change in use of the terms ‘competency’ and ‘competence’ in the existing literature. Nehrir et al. (2016) [34] identified that the competence definitions can be varied in a variety of ways based on profession and country. However, Fukada (2018) [9] contrasts the earlier discussion, having illustrated that the competence definitions were established from the previous literature according to international standards and the literature used in international and local databases. A range of variations has already been identified in the literature regarding the definitions of competence [9,34,35,36]; however, there is a consensus that competence encompasses a wide range of intricate qualities, including knowledge and critical thinking, practical and behavioural skills (like the ability to complete a task safely and successfully), and personal and professional qualities (like moral behaviour and values).

Assessment tools for nursing competence are diverse, covering various domains such as professional attributes and critical thinking. Although these tools are developed in various countries, there is a consensus that these tools are effective in many domains, including professional attributes, ethical practices, communication and interpersonal relationships, nursing processes, critical thinking and reasoning. The tools were designed and evaluated to achieve the needs of the assessment competence for a single institution according to a specific case measured in each situation and context, as well as the purpose of the assessment [41,42,43,50,51]. However, no universally applicable method exists, with tools tailored to specific contexts and purposes. Therefore, it is observed that there is a variety in nursing assessment competence methods and tools among countries and higher education institutions [17,22,23,37,63]. 

Reliability and validity concerns persist, highlighting the need for standardized assessment instruments aligned with professional standards [41,42,43,45,48,49]. Ko and Yu (2019) [51] indicated that a poor content validity in studies was described in the assessment tool. This finding is contrary to Charette et al. (2020) [39], who found that there is insufficient proof on the reliability and validity of the competence tools. Ossenberg et al. (2016) [47] discussed that the validity and reliability of the Australian Nursing Standards Assessment Tool (ANSAT) needed to be assessed.

The nursing profession is an internationally recognised professional qualification. So, it would be better to establish assessment competence tools for nursing students based on agreed professional standards for ensuring the abilities of nursing students in providing safe nursing care [20]. Standardized assessment tools would facilitate comparability and transparency across healthcare settings globally, enhancing nursing graduates’ readiness for practice.

As previously stated, the majority of preceptors from different studies face challenges in converting competence documentation into measurable criteria such as knowledge, abilities, and attitudes. This was ascribed to difficulties comprehending the terminology in the competency statements, indicating that the competency assessment document should be reviewed, as should the requirement for a trained clinical guide to assist preceptors in their duties. Consequently, it is important for people who are setting up or reviewing a nursing program to consider the principles and the mechanism of assessment competence; to establish a valid and reliable clinical assessment tool to assess certain levels of competence of students, it has to be written in simple, understandable and clear language, distinguishing between various levels of competence. Assessment competence tools should be developed collaboratively between clinical and academic colleagues, and training and other support (such as continuous mentorship) also need to be put in place to enable both preceptors and nursing students to clearly understand the criteria in the tool. Also, support for preceptors should be provided to enhance the quality of assessment process and achieve students’ outcomes to a high standard.

### 4.1. Strengths

This review has explored assessment competence in nursing clinical practice from various aspects, including definitions of competency, tools used to assess nursing students’ competence, and the viewpoints of nursing preceptors in relation to assessment of competence. Therefore, the scoping review would cover a wide range of areas in assessment competence. Although only English language studies were included in this review, the work represents 19 countries over the past 20 years, and thus, includes an in-depth review of the literature in this area.

### 4.2. Limitation

This review has a limitation in that it only included studies published in English.

## 5. Conclusions

The results of this scoping review can be used by nurse educators to help in facilitating the competency assessment process in clinical practice from various aspects. This review provides a range of competency definitions in the nursing field and a multitude of the quantitative instruments available to assess clinical nursing students’ competence from many different countries. Further, it is important that the views of preceptors are considered because of their significant role in assessment the competency process in clinical practice with nursing students, as well as when designing tools for the assessment of competency, because assessment is critical to gather information about learning and measuring performance, which can be used to confirm the outcome and competency among nursing students, and also determines their eligibility to be placed on a professional nursing register. This would contribute to reducing the complexity of the assessment competency faced by nursing educators and preceptors worldwide.

## Figures and Tables

**Figure 1 healthcare-12-01031-f001:**
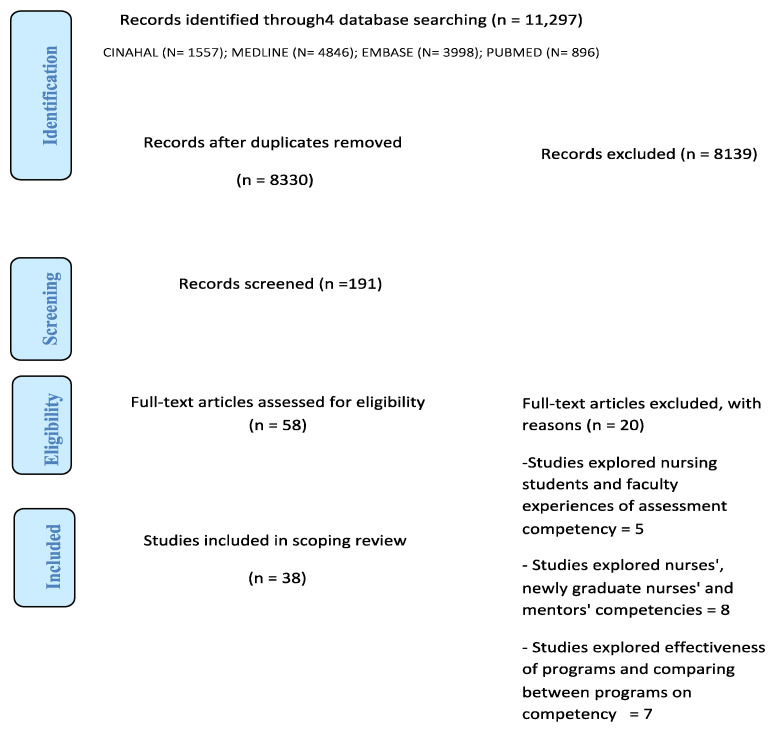
PRISMA Flow Diagram [33].

**Table 1 healthcare-12-01031-t001:** The PCC model.

Acronym and Components	Description of Components
Population (P):	Nursing Preceptors who are responsible for assessing nursing students
Concepts (C):	Competency and Assessment Competence
Context (C):	Clinical area in nursing

**Table 2 healthcare-12-01031-t002:** Terms and the relevant synonyms.

Terms	The Relevant Synonyms
‘Definition of Competence’:	Proficiencies; Efficiencies; Academic progression; Academic performance; Academic achievement; Competency assessment; Professional competence; Clinical competence; Clinical assessment tools; Definition of competence; Concept of competence; Competency tools.
‘Preceptor’:	Educator; Health educator; Mentor; Supervisor; Student supervision; Practice assessor; Lecture; Nursing academic.
‘Nursing’:	Nurse; Nurses; Nursing (Nurs*).

**Table 3 healthcare-12-01031-t003:** Search with descriptor and Boolean AND/OR operators.

Descriptor	Search with Descriptor and Boolean AND/OR
‘Competency’:	Proficiencies OR Efficiencies OR Academic progression OR Academic performance OR Academic achievement OR Competency assessment OR Professional competence OR Clinical competence OR Clinical assessment tools.
	AND
‘Preceptor’:	Educator OR Health educator OR Mentor OR Supervisor OR Student supervision OR Practice assessor OR Lecture OR Nursing academic.
	AND
‘Nursing’:	Nurse OR Nurses OR Nursing (Nurs*).

**Table 4 healthcare-12-01031-t004:** (**1**) Definitions of competence (4 studies). (**2**) Identification of existing tools OR specific tools that are developed for assessing competencies and Evaluation of specific tools that are developed for measuring clinical nursing assessment competency (18 studies). (**3**) Preceptors and Mentors Viewpoints (16 studies).

(1)						
Author Year Country	Aim of the Study	Study Design	Key Words	NO. of Studies OR Sample	Quality Appraisal and Tool Used	Result
Nehrir et al. (2016) [34] Iran	To explore the definition, domains, and levels of nursing student’s competency.	Systematic review	Competency, Nursing Student, Systematic Review	20 studies	---------	Nursing students’ competency encompasses individual experiences, dynamic processes, and positive social changes in professional life, affecting meta-cognitive abilities, motivation, decision making, job involvement, professional authority, self-confidence, and knowledge. Dominos of nursing students’ competency are educational, cultural, individual, professional and inter-professional, research, clinical and practical domains. There are seven nursing student competency levels that were identified.
Windsor et al. (2012) [35] Australia	To develop an analysis of competency-based clinical assessment of nursing students across a Bachelor of Nursing degree course.	Review	Competence, competency, Soft skills, Nursing, Professionalisation.	406 clinical assessment tools from years 1992–2009	------------	The existence of a hierarchy of competencies that prioritises soft skills over intellectual and technical skills; the appearance of skills as personal qualities or individual attributes; and the absence of context in assessment.
Fukada (2018) [9] Japan	To review the research on definitions and attributes of nursing competency in Japan as well as competency structure, its elements and evaluation. Furthermore, to investigate training methods to teach nursing competency.	Review	Concept, Development support, Nursing competence, Structure of dimensions			Nursing competency is crucial for fulfilling nursing responsibilities and establishing a foundation for education. However, the concept has not been fully developed, leading to challenges in defining and structuring competency levels, training methods, and improving nursing quality.
Mrayyan et al. (2023) [36] Jorden	To clarify the concept of competency in nursing practice and propose an accurate definition.	Review	Competency, Competence	10 studies	Walker and Avant’s approach	Competency in nursing practice, characterized by knowledge, self-assessment, and dynamic state, has been reported to improve patient, nurse, and organizational outcomes.
**(2)**						
**Author Year** **Country**	**Aim**	**Study Design**	**Key Words**	**NO. of Studies OR Sample**	**Quality Appraisal Completed and Tool Used**	**Result**
Ličen and Plazar (2015) [24] Slovenia	To identify existing tools that purport to measure clinical nursing competence through the use of a systematic literature review to consider the possibilities of using them in nursing education in Slovenia.	Systematic literature review	Nursing, Competencies, Assessment, Education	7 studies	PRISMA guidelines	○ The availability of highly reliable tools ○ Tools enable assessment of clinical competences ○ Lack of clarity of some competencies students must achieve.
Yanhua and Watson (2011) [37] China	To investigate trends in the evaluation of clinical competence in nursing students and newly qualified nurses over the last 10 years.	Literature review	Competence, Assessment, Nursing	23 studies	A literature review following PRISMA guidelines	○ Instrument development and testing (n = 4) ○ Approaches to testing competence (n = 7) ○ Assessment and related factors (n = 12).
Immonen et al. (2019) [22] Finland	To identify the current best evidence on the assessment of nursing students’ competence in clinical practice.	Systematic review	Assessment, Clinical practice, Evaluation, Nursing student, Systematic review	6 studies	critical appraisal	Assessment tools for nursing competence typically focus on professional attributes, ethical practices, communication, and critical thinking. Clinical learning environments and mentoring support students’ learning, ensuring objectivity and reliability.
Charette et al. (2020) [38] Australia and Canada	To appraise and synthesize evidence of empirical studies reporting assessment of new graduate nurses’ clinical competence in clinical settings.	Mixed methods systematic review	Assessment, Clinical competence, Competency assessment, Literature review, Mixed methods, New graduate nurse, Nursing, Systematic review	42 studies	Mixed Methods Appraisal Tool framework for quality appraisal	New graduate nurses show good competence, with significant increases from 0–6 months, but inconsistent from 6–12 months. Quantitative tools need reviewing for rigour.
Charette et al. (2020) [39] Australia and Canada	To analyse, evaluate and synthesize the measurement properties of scales used to assess new graduate nurses’ clinical competence in clinical settings.	Systematic psychometric review	Clinical competence, Competence assessment, New graduate nurse, Nursing, Psychometric properties, Scale, Systematic review	Ten scales	Consensus-based standards for the selection of health Measurement Instruments (COSMIN) methods.	There is little evidence on the measurement properties for each scale regarding their validity and reliability; responsiveness was not assessed for any scale. Every scale evaluated in this review had different characteristics (length, subscales, response options). Therefore, selection of the most appropriate scale depends on the context and purpose of the assessment.
Van Horn and Lewallen (2023) [40] USA	To examine the research literature to identify objective, replicable measurement of clinical competence in undergraduate nursing education.	A comprehensive search	Clinical Evaluation, Instruments, Nurse Competence, Nursing Education	Twelve reports	PRISMA	The study utilized various measures to assess competence, including knowledge attributes, attitudes, behaviors, ethics, personal attributes, and cognitive or psychomotor skills, primarily utilizing researcher-created instruments.
Laokhompruttajarn et al. (2021) [41] Thailand	To develop a professional competency evaluation model of nursing students.	Research and development process	Development, Evaluation Model, Professional Competency, Nursing Students	A research and development process divided into four steps	The research and development process divided into four steps	The model identifies seven core and four functional competencies for nursing students, demonstrating discriminant validity and high levels of feasibility, appropriateness, accuracy, and usefulness in evaluating professional competencies.
Manz et al. (2022) [42] USA	To provide a review of the literature associated with the C-SEI and the C-CEI and to lay the foundation for the upcoming revision of the instrument consistent with the updated AACN Essentials (2021).	Review of the literature	C-CEI, C-SEI, Competency, Assessment, Evaluation	40 studies	PRISMA	The C-SEI and C-CEI are valid and reliable instruments used to evaluate students, graduate nurses, and professional nurses in clinical and simulated learning environments. They have been adapted for an interprofessional competence assessment and are essential for competency-based nursing education.
Sastre-Fullana et al. (2017) [43] Spain	To describe the development and clinimetric validation of the Advanced Practice Nursing Competency Assessment Instrument (APNCAI) through several evidence sources about reliability and validity in the Spanish context.	Develop the APNCAI tool (literature review and instrument content consensus)	---------	600 nurses	APNCAI development was based on a multisequential and systematic process: literature review, instrument content consensus	The eight-factor competency assessment latent model, APNCAI, is suitable for APN competency assessment in Spain, with adequate reliability and validity, making it useful for healthcare policy programs.
Sahin et al. (2021) [44] Turkey	To evaluate the education/learning outcomes of nursing students’ competence using the nursing student competency instrument in a Turkish sample.	Descriptive cross-sectional quantitative study	Nursing Students Competence Instrument (NSCI), Student nurse, Nursing	390 nursing students.	Nursing Students Competence Instrument (NSCI)	Nursing students exhibit high self-perceived competence in leading humanity concerns and advancing career talents, but low competence in dealing with tension is observed.
O’Connor et al. (2009) [45] Ireland	To describe a collaborative project conducted by the three principal universities in Dublin to implement and evaluate a competence assessment tool for use by nursing students and their assessors while on clinical placements.	Development of the assessment tool, and Evaluation	Competence, Clinical assessment, Ireland Collaboration	27 preceptors	Shared Specialist Placement Document (SSPD) tool	The evaluation provides a promising foundation for developing competence-based assessment strategies for nursing in Ireland, but further work is needed on preparation and support for assessors and students.
Zasadny and Bull (2015) [46] Australia’s island state Tasmania- Australia	The ASAP model was evaluated by gathering clinical facilitator and student feedback over two 13 week semesters during practice and formal meetings, as well as review of student performance data.	Developing the ASAP tool through formal education sessions	Assessment Competence Undergraduate nursing, Health, Clinical practice	225 final year nursing students.	Amalgamated Student Assessment in Practice (ASAP) model and tool	The ASAP model functioned effectively as an assessment tool, focused diagnostic tool, removal from Professional Experience Placement (PEP) support tool and a framework for documenting evidence.
Ossenberg et al. (2016) [47] Australia	To advance the assessment properties of a new instrument, the Australian Nursing Standards Assessment Tool (ANSAT), and investigate the acceptability of this instrument when applied to the evaluation of the professional competence of nursing students in authentic practice settings.	Validation study of ANSAT	Assessment, Work-based Instrument, Valid, Professional standards, Performance, Undergraduate, Nursing student	23 clinical assessors	Australian Nursing Standards Assessment Tool ANSAT instrument	The pilot study supports the ANSAT instrument, recommending testing on a larger cohort for generalizability. The instrument, with supportive behavioral cues, enables clear, consistent, and collaborative workplace-based assessment.
Hwang et al. (2018) [48] Taiwan	To develop a tool for measuring competency in conducting health education and to evaluate its psychometric properties in a population of entry-level nurses.	A cross-sectional survey	Competency, Health education, Psychometric, Scale development	457 nursing students and 165 clinical nurses	Health Education Competency Scale (HECS) developed in this study	The Health Education Competency Scale, a four-factor solution, accounted for 75.9% of the variance in entry-level nurses competency, with good reliabilities and construct validity.
Kim and Shin (2022) [49] Korea	To develop a scale to assess nursing practice readiness and verify its validity and reliability.	Development of the Nursing Practice Readiness Scale and testing its validity and reliability	Education, Nursing, Instrument validation, New graduate nurses, Psychological test	430 new graduate nurses	development of the Nursing Practice Readiness Scale	The Nursing Practice Readiness Scale, consisting of 35 items, assesses clinical judgment, nursing performance, professional attitudes, patient-centeredness, self-regulation, and collaborative interpersonal relationships, ensuring adequate model fit.
Gardulf et al. (2016) [50] Sweden	To investigate self-reported competence among nursing students on the point of graduation (NSPGs), using the Nurse Professional Competence (NPC) Scale, and to relate the findings to background factors.	Development the scale	Nurses’ competence, Professional nursing, Nursing education, Nursing students, Graduate nurses, Quality and Safety in care, NPC Scale	1086 nursing students on the point of graduation (NSPGs)	Nurse Professional Competence (NPC) Scale,	Nursing Students on the Point of Graduation (NSPGs) were highest for the four CAs connected with patient related nursing and lowest for CAs relating to organisation and development of nursing care. The Nurse Professional Competence (NPC) scale can be used to identify and measure aspects of self-reported competence among NSPGs.
Ko and Yu (2019) [51] South Korea	To develop a competency assessment instrument for nurses who have completed an outcome-based educational program based on national standards and to assess the content-, construct-, and criterion-related validity of that instrument.	Development and evaluation the tool	advanced beginner nurses, competency, nursing graduates, outcome-based education.	141 nurses with 1–3 years’ clinical experience	The construct- and criterion-related validity of the nursing core competency assessment tool	A competency assessment instrument for nurses with 1–3% experience was developed, addressing nursing research, policy awareness, and leadership competencies. However, content validity issues led to the removal of these competencies for college graduates.
Huang et al. (2022) [52] Taiwan	To develop and validate a nursing competence instrument for nursing students in bachelor training.	descriptive and explorative study design	Nursing students, Competence, Reliability, Validity, Instrument development	241 nursing students	Nurse Competence Scale	The tool demonstrated satisfactory psychometric qualities, making it an invaluable resource for assessing nursing students’ proficiency during their bachelor’s programme.
**(3)**						
**Author and Year** **Country**	**Aim**	**Study Design**	**Search Strategy**	**Number of Studies/OR Sample**	**Quality Appraisal Completed and Tool Used**	**Result**
Butler et al. (2011) [53] Ireland	To explore preceptors’ perspectives concerning the content of a competency assessment tool and experience of the competency assessment process in the disciplines of general, mental health and intellectual disability nursing in the Mid-West region in Ireland.	Mixed methods design	Competency assessment, Preceptors, Student nurses	837 preceptors	A specifically designed questionnaire was developed	Preceptors found difficulties understanding the used language in the document of the assessing competence.
Cassidy et al. (2012) [12] Ireland	To evaluate clinical competence assessment in BSc nursing registration education programmes.	Mixed methods design	Competency, Assessment, Preceptors, Undergraduate, Nursing, Qualitative	16 preceptors	Focus groups	The result of the study indicated three categories emerged of the preceptors: (1) attitudes to competencies, (2) being a preceptor, (3) competencies in practice. Also, competing demands influenced preceptors’ assessment experiences in the clinical environment. Preceptors found some difficulties such as: understanding the used language in the document of the assessing competence, and integration of skills into the assessment.
Fahy et al. (2011) [54] Ireland	To evaluate clinical competence assessment in pre-registration BSc nursing programmes in one geographical area in the Republic of Ireland.	Mixed methods design	Assessment, Clinical skills, Competence, Education, Preceptors, Student nurses	Phase 1: 13 students and 16 preceptors. Phase 2: 232 students and 837 preceptors	Focus groups and developed questionnaires	Preceptors struggled with the language used in the competence assessment document, stating it was overly broad and ambiguous. They emphasized knowledge of clinical skills, highlighting the need for user-friendly language and a comprehensive assessment of all aspects.
Wu et al. (2017) [2] Singapore	To explore the perceptions of clinical nurse leaders and academics on clinical assessment for undergraduate nursing education during transition to practice.	Explorative qualitative approach	Academics, Clinical assessment, Clinical nurse leaders, Clinical nursing education, Nurse preceptors, Undergraduate nursing students.	6 clinical nurse educators, and 8 academics	A semi-structured interview	Clinical Nurse Leaders identified four key themes during clinical assessment: the need for a reliable tool, preceptor competence issues, challenges faced by nursing students, and the need for collaboration between clinical and academic sectors to support preceptors and students.
Wu et al. (2016) [18] Singapore	To explore the perspectives of preceptors about clinical assessment for undergraduate nursing students in transition to practice	Exploratory qualitative approach	Clinical Assessment, Clinical Guidance, Clinical Nursing Education, Feedback, Nurse Preceptors, Nursing Education	17 preceptors	Focus group discussion.	Preceptors reported five themes about clinical assessment in transition to practice which were: 1. The need for a valid and reliable clinical assessment tool. 2. Meaningful reflection and feedback. 3. Varied modes in clinical assessment. 4. High level of commitment and struggles with dual roles. 5. The need to enhance the support system for preceptors.
Kennedy and Chesser-Smyth (2017) [55] Ireland	To explore the lived experiences of the preceptors during the assessment process using a phenomenological approach	Qualitative study	Nursing students, Clinical assessment, Clinical competence, Preceptors and clinical assessment	9 preceptors from two clinical sites.	Individual in-depth interviews	The preceptors discussed their experiences with the assessment process, including first impressions, emotional turmoil, and workplace demands. They suggested a tripartite approach for enhanced decision-making, enhancing objectivity and reducing emotional turmoil in cases of incompetence or borderline competence.
Burke et al. (2016) [56] Ireland	To explore Irish preceptors’ experience of using a competence tool to assess undergraduate nursing students’ clinical competence.	Mixed methods design	Preceptors, Competence, Assessment tool, Undergraduate, nursing students	Phase 1: (17 preceptor) Phase 2: 843 preceptors	Mixed methods design	The preceptors indicated these themes of their experiences of using a competence tool which were: (1) Challenges of using the assessment competency tool, especially the complexity of the language (2) Valuing adult learners and recognising competence.
Cassidy et al. (2017) [57] UK	To explore mentors’ experiences of assessing nursing students on the borderline of achievement of competence in clinical practice and to develop a substantive theoretical explanation of this phenomenon.	Grounded theory qualitative study	Assessment, Borderline decision-making, Clinical practice, Competence, Grounded theory, Mentor, Nurse education, Nursing, Qualitative, student	Phase 1: 20 mentors Phase 2: 8 individual semi-structured interviews and 7 focus groups with mentors and 38 practice educators	Phase one: semi structured interviews Phase two: semi-structured interviews and focus groups	The results of this study reported three categories from mentors which were: (1) ‘the conundrum of practice competence’, (2) ‘the intensity of nurturing hopefulness’, and (3) ‘managing assessment impasse’. This conundrum in defining the form of competency, the level of assessment was complex because it assessed students’ ability for reflection and thinking critically. Mentors have shown competence is not merely following directions but rather interpreting and responding to changing contexts in practicing.
Helminen et al. (2017) [25] Finland	To describe the phenomenon of final assessment of the clinical practice of nursing students and to examine whether there were differences in assessments by the students and their teachers and mentors.	Descriptive cross-sectional design.	Clinical practice, Final assessment, Mentors, Nurse educator, Nursing education, Nursing students	276 nursing students, 108 their teachers and 225 mentors.	The questionnaire was developed for this study by the authors based on a literature review of previous research.	Results showed four main factors that relate to nursing students’ final assessment: ‘Fair and consistent assessment given by mentors’; ‘Criteria based on honest and direct assessment’; ‘Assessment taking into account multiprofessional views’; ‘Teachers’ presence in the assessment situation’.
Almalkawi et al. (2018) [21] UK	To evaluate the empirical and theoretical literature on the challenges mentors face in interpreting and assessing levels of competence of student nurses in clinical practice.	Integrative review	Integrative review, Students, Mentors, Practice-based assessment, Competence, Interpretation, Feedback, Rubric	8 records	Mixed Methods Appraisal Tool	The results reported that there are difficulties in the language used for describing competencies. There is a challenge in distinguishing among competency levels. There is a lack of constructive and clear feedback to nursing students. Lack of transparent and explicit criteria hinders accurate and fair assessment of students
McCarthy and Murphy (2008) [27] Ireland	To explore to what extent preceptor nurses use the devised assessment strategies to clinically assess BSc students in one university in The Republic of Ireland.	Quantitative approach with a qualitative dimension	Preceptor, Clinical assessment, Nursing students, Assessment strategies	470	Questionnaires for Psychiatric and Intellectual Disability nursing preceptors	Numerous preceptors were inexperienced, did not completely understand the assessment procedure, and did not use all the required assessment techniques while assessing students in clinical practice.
Burden et al. (2018) [58] UK	To investigate how mentors form judgements and reach summative assessment decisions regarding student competence in practice	Two-stage sequential embedded mixed-methods design.	Assessment, Competence, Decision-making, Judgements, Mentors, Mixed-methods, Nursing, Practice-based assessors, Student	Stage 1: mentor (N = 330 from 270 mentors) Stage 2: mentors (N = 17).	(NMC) Practice Assessment Documents (PADs)	The study suggests that assessment documentation and strategies do not significantly influence mentor judgements and decisions, but decision-making theory can help understand assessment competence and clarify variability in mentor decisions.
Hughes et al. (2019) [59] Australia	To describe both tertiary and industry-based assessors’ experiences of grading nursing student performances in clinical courses when that performance was not a clear pass or fail.	A pilot study using a descriptive survey design	Failure to fail, Fitness for practice, Competence assessment, Descriptive survey design	149 assessors	The survey was developed	Assessor had a clear duty for patient care and the nursing profession. However, 23.5% of assessors gave student performance the benefit of the doubt. They claimed failing student performances while also reporting passing nursing students.
Brown and Crookes (2017) [60] Australia	To generate a series of guidance notes by asking experienced nurses to explain how they assessed the competency level of nursing students	Modified nominal group (Consensus methodology)	Student nurse competence, Competence assessment, Consensus methodology	Groups were facilitated across 7 of the eight states and territories in Australia.	A modified nominal group technique (NGT) was used in order to elicit expert opinion from the clinicians	Guidance notes were developed for the assessor and the student. These guidance notes demonstrate what is expected of the nursing student related to illustrating their competence; therefore, the assessor can assess the competence of the student, using specific guidance for supporting them.
Nugent et al. (2020) [61] Ireland	To gain a better understanding of the preceptors’ decision-making process when nursing students’ competence is below required standards, and identify the perceived barriers and enablers supporting them in this task.	Descriptive quantitative approach.	Clinical competence, Failure to fail, Nursing students, Preceptorship, Underperforming nursing student	1530 preceptors were invited to participate: 365 valid questionnaires were returned	Developed questionnaire.	Preceptors were enjoying their role as assessors and working with students and getting positive feedback from students. However, preceptors ask for more support related to assessment documents from fellows and further training in relation to providing negative feedback to students
Borren et al. (2023) [62] Newzealand	To identify current competence assessment practice and determine how competence assessment is constructed in order to reflect student development.	Qualitative exploratory-descriptive design.	Students, Nursing, Education, Baccalaureate, Clinical competence, Clinical assessment, Clinical education	10 Nurse educators	Semi-structured interviews	Three themes emerge: clinical assessment pedagogy, measure of competence, and relational assessment practice. The process of performing competency assessments varied significantly between and within institutions, and it was noted as a problem to scaffold these assessments throughout the degree curriculum.

**Table 5 healthcare-12-01031-t005:** Summary of the countries in which the research took place.

Publication Year	2008–2023	No.	Included Studies
Country	Ireland	8	O’Connor et al. (2009) [45]; Butler et al. (2011) [53]; Cassidy et al. (2012) [12]; Fahy et al. (2011) [54]; Kennedy and Chesser-Smyth (2017) [55]; Burke et al. (2016) [56]; McCarthy and Murphy (2008) [27]; Nugent et al. (2020) [61]
	Australia	6	Windsor et al. (2012) [35]; Zasadny and Bull (2015) [46]; Ossenberg et al. (2016) [47]; Hughes et al. (2019) [59]; Brown and Crookes (2017) [60]
	United Kingdom	3	Cassidy et al. (2017) [57]; Almalkawi et al. (2018) [21], Burden et al. (2018) [58]
	Finland	2	Immonen et al. (2019) [22]; Helminen et al. (2017) [25]
	Singapore	2	Wu et al. (2017) [2]; Wu et al. (2016) [18]
	Korea	2	Kim and Shin (2022) [49]; Ko and Yu (2019) [51]
	Taiwan	2	Hwang et al. (2018) [48]; Huang et al. (2022) [52]
	USA	2	Manz et al. (2022) [42]; Van Horn and Lewallen (2023) [40]
	Sweden	1	Gardulf et al. (2016) [50]
	Spain	1	Sastre-Fullana et al. (2017) [43]
	Slovenia	1	Ličen and Plazar (2015) [24]
	Turkey	1	Sahin et al. (2021) [44]
	Japan	1	Fukada (2018) [9]
	China	1	Yanhua and Watson (2011) [37]
	Thailand	1	Laokhompruttajarn et al. (2021) [41]
	Jorden	1	Mrayyan et al. (2023) [36]
	Iran	1	Nehrir et al. (2016) [34]
	New Zealand	1	Borren et al. (2023) [62]
	both Australia and Canada	2	Charette et al. (2020) [38]; Charette et al. (2020) [39]
Research Design	Quantitative Studies	15	Laokhompruttajarn et al. (2021) [41]; Sastre-Fullana et al. (2017) [43]; Sahin et al. (2021) [44]; O’Connor et al. (2009) [45]; Zasadny and Bull (2015) [46]; Ossenberg et al. (2016) [47], Hwang et al. (2018) [48]; Kim and Shin (2022) [49]; Gardulf et al. (2016) [50]; Ko and Yu (2019) [51]; Helminen et al. (2017) [25]; McCarthy and Murphy (2008) [27]; Hughes et al. (2019) [59]; Nugent et al. (2020) [61]; Huang et al. (2022) [52];
	Qualitative Studies	6	Wu et al. (2017) [2]; Wu et al. (2016) [18]; Kennedy and Chesser-Smyth (2017) [55]; Cassidy et al. (2017) [57]; Brown and Crookes (2017) [60]; Borren et al. (2023) [62];
	Mixed Methods Studies	5	Butler et al. (2011) [53]; Cassidy et al. (2012) [12]; Fahy et al. (2011) [54]; Burke et al. (2016) [56]; Burden et al. (2018) [58];
	Reviews	12	Nehrir et al. (2016) [34]; Windsor et al. (2012) [35]; Fukada (2018) [9]; Ličen and Plazar (2015) [24]; Yanhua and Watson (2011) [37]; Immonen et al. (2019) [22]; Charette et al. (2020) [38]; Charette et al. (2020) [39]; Manz et al. (2022) [42]; Almalkawi et al. (2018); Van Horn and Lewallen (2023); Mrayyan et al. (2023) [36].
Sample	Preceptors	7	Butler et al. (2011) [56]; Cassidy et al. (2012) [12]; Fahy et al. (2011) [54]; Wu et al. (2016) [18]; Kennedy and Chesser-Smyth (2017) [55]; Burke et al. (2016) [56]; McCarthy and Murphy (2008) [27]; Nugent et al. (2020) [61];
	Nurses	3	Sastre-Fullana et al. (2017) [43]; Hwang et al. (2018) [48]; Kim and Shin (2022) [49];
	Mentor	3	Cassidy et al. (2017) [57]; Helminen et al. (2017) [25]; Burden et al. (2018) [58];
	Clinical Assessors	3	Ossenberg et al. (2016) [47]; Hughes et al. (2019) [59]; Brown and Crookes (2017) [60];
	Clinical Educators	2	Wu et al. (2017) [2]; Borren et al. (2023) [62];
	Nursing teachers	1	Helminen et al. (2017) [25];
	Practice educators	1	Cassidy et al. (2017) [57];
	Nursing students	8	Sahin et al. (2021) [44]; O’Connor et al. (2009) [45]; Zasadny and Bull (2015) [46]; Hwang et al. (2018) [48]; Gardulf et al. (2016) [50]; Ko and Yu (2019) [51]; Fahy et al. (2011) [54]; Helminen et al. (2017) [25]; Brown and Crookes (2017) [60]; Huang et al. (2022) [52].

## Data Availability

Data are contained within the article.

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
