# Peer review of "Assessing the Competence of Nursing Students in Clinical Practice: The Clinical Preceptors’ Perspective"

_healthcare, 2024, doi:10.3390/healthcare12101031_

Round 1

Reviewer 1 Report

Comments and Suggestions for Authors

Dear authors,

 I was pleased to review your article. I find the proposed topic very interesting and valuable for improving the practical education of the nursing profession. The introduction section provides valuable insights into the topic of nursing education. The methodology is well written and the PRISMA algorithm demonstrates the value of the study conducted. The Discussion section and the Conclusion are well executed and written.

I have some suggestions:

 -The paper is 22 pages long. I suggest condensing tables and shortening sections to reduce the overall page count.

 - In the Methodology section, what was the reason for choosing this methodology and why were these databases selected? Short communication, letter to the editor and brief article, or lack of access to the full text of the articles, how was it managed?

 - I recommend adding a separate section on limitations and strengths of the study.

 - I recommend checking the Bibliography section because it does not follow the format of the journal.

 I hope this helps

Author Response

Thank you son much. 

Please find attached. it is response to Reviewer's 1 comments. 

Reviewer 2 Report

Comments and Suggestions for Authors

Assessing the competence of nursing students in clinical practice is a critical aspect of their education.

Assessment tools used to evaluate nursing students’ competence commonly focus on various domains, including professional attributes, ethical practices, communication, nursing processes, critical thinking, and reasoning. However, there is inconsistency among assessment methods and tools across different countries and institutions. Reliable and valid instruments are essential for accurate assessment, but achieving consistency seems to be an ongoing need. Furthermore, preceptors find assessing students’ competence particularly challenging. It seems, that the nursing education community still needs to work toward consistent and evidence-based assessment practices. Collaboration among educators, preceptors, and students is crucial to ensure that assessments are fair, reliable, and aligned with professional standards.

I think the article could be published as is. The only change I would like is that the title of the article be changed to clearly describe the assessment of the competence of nursing students from the point of view of clinical supervisors.

Additional comments:

The article is written smoothly and progresses logically. The topic is very familiar to me as a professional nursing educator and researcher, thus the content of clinical evaluation is difficult in many times in reality as well.

The use of concepts is consistent. The description of the procedure is sufficient and concise.

Based on the description of the method, you can understand how the analysis was done and how the results were arrived at.

The scoping review seems to have been done carefully following the selected inclusion and exclusion criteria.

The sources/ references used are appropriate.

The article title does not match the content. I urge the working group to change that.

No any other changes needed on my behalf.

Author Response

Thank you for your comments.

Please find attached. It is the response to Reviewer's 2 comments. 

Reviewer 3 Report

Comments and Suggestions for Authors

Dear authors. Thank you for providing me with the opportunity to review your paper titled "Current Thinking on Clinical Assessment for Nursing – a scoping review." I found it to be an engaging and pertinent topic relevant for the journal. Exploring central elements of nursing education is crucial for the enduring strength of our profession, grounded in research and evidence. Therefore, I commend your dedicated efforts in this regard.

A brief summary.

The aim of this scoping summary was to explore and synthesise the existing evidence regarding the assessment of competence and assessment methods and the preparedness of nursing preceptors to assess the competency of nursing students in clinical practice, having three research questions: 1. What are the definitions of competence? 2. What are tools used for assessing competence? 3. What are the viewpoints of nursing preceptors and mentors regarding the assessment of competence?

The study followed the methodological framework of Arksey and O'Malley (2005) and further by the advances made by Levac et al 2010.

The search was conducted up to June 2022 in CINAHL, Ovid Medline, Embase, and PubMed databases.

From 8237 hits for the searches, the authors end up with 34 included studies. The search is illustrated in a Prisma flow diagram. 

The included studies are presented in three ways, first in a descriptive way, second in a tabular way, and last, in short presentations regarding ‘definitions of competencies, Tools, and viewpoints of nursing preceptors. The study has a discussion and ends up with a section on the conclusion.

General concept comments

I find the review of this topic of most importance. However, I think that the manuscript requires some clarification. I hope that my review can shed light on some elements that might be taken under consideration for improvement. Know that I acknowledge your large work.

The article has severe weaknesses due to the coherence between the method stated and the work done. I don’t think that this version of your manuscript is ready for publication yet.  Especially I found that there seems to be some confusion about the method of Scoping review. You write: “The aim of the scoping review is to explore and synthesise the existing evidence…”  However, it seems not to be a synthesis. - rather at descriptive presentation. Further, there is very little information about the searches in the database. For example, how many hits for each database. I will also expect more transparency regard MesH terms heading and text words used in the search to be able to evaluate the rigor of the search. Is also very unclear how the results occurred. There are no descriptions or examples of the analysis. Further, there is no discussion of your results – the section of discussion seems to be a continuation of the section of the descriptive result section. No new literature or theory is used to discuss your findings. This makes the paper very weak regarding the cohesion between the method stated and the actual work reported in the paper.

Review: commenting on the completeness of the review topic covered, the relevance of the review topic, the gap in knowledge identified, the appropriateness of references, etc.

I acknowledge your large work on the searches and the number of papers included. However, some work is needed to reach a more rigorous level of your scientific work.

Line 2: Consider the title. The title does not cover all your topics in the review – which might exclude the review from being identified by healthcare professionals interested in for example assessment tools or definitions.

Line 52: an error in the typing

Line 70:  an error in the typing

Line 88-89: the sentences regarding PROSPERO might be more relevant in the section on method. Be also aware that PROSPER doesn’t accept scoping reviews anymore.

Line 96: the method states summarising and reporting – however in line 85 you state that you want to synthesise. There must be coherence between the aim and the method used.  

Line 101-103: you state to have the research questions. I need some clarification on how these questions are developed. Further, I think that it is three very large questions which each might have it own review to cover each very important question in a more in-depth way.

Line 105: you write “An analysis of the keywords was carried out….” It is unclear how you made the analyses. I think more transparency is needed.

Line 110: table 1. I think that the table requires details. For example, which MeSH terms, Headings and/or text words did you use regarding each database? How did you combine the words in the systematic search? Did you use the Boolean operators? Further, I wondered why the word “definition” is not represented and the word “Tool” only occurs once - and no synonyms are represented in the search words.  

Line 113: a search done in June 2022 – it is almost two years ago – I will expect a refreshing of the search – ensuring that no new or important studies have been published.  Further, I am wondering why the database ERIC (Education Resources Information Center) is a database focusing on education.

Line 124: I need argumentation for the twenty-year limitation.  Also, why include both systematic review and primary studies?

Line 129-138: I need much more transparency in the analysis process. How did you make the syntheses (which I don’t think you have made)? Why use Joanna Briggs's tool when drawing on Arksey and O'Malley – consider methodical rigor. At least argue why you used the JBI tool. Further, if you mean that you are making a synthesis – I need information regarding the quality assessment of each paper included in your review.

Line 150: Your review will benefit from providing more information regarding the screening process. Provide information regarding hits for each database. I also need information on how you have explored the included reviews for duplicates related to the included primary studies in your scoping review.

Line 168: table 2. A very comprehensive table – however consider how you can make the collum of results more alike. It will improve the reading and comprehension of the relation between your included studies and your results.

Line 183: I need an argumentation of how Table 3 improves the insight of the included studies – you have reported it in section 3.1.1. I don’t understand why the origin of the study, method, and samples is so much important – especially as you don’t discuss this result or conclude on the elements.  

Line 196 – 247: These three results are lacking the elements of a synthesis. The sections do not cover what may be expected for reporting the results of a review. The content of these three paragraphs is not sufficient in a scientific article. I believe these paragraphs particularly demonstrate that this work is premature.

Line 248: First of all – once again you state that this is a synthesis- I wonder why you don’t reflect on your method and the result reported.

Line 315-332: The section of the conclusion seems to make a statement on implication for practice rather than conclude on the comprehensive study.   

The discussion lacks the elements of scientific perspectives and/or theory challenging your results. Also, there is no discussion of the method and strengths and weaknesses of your study.  I think it is very important to discuss our scientific work.

As such, due to this review, I cannot recommend that the paper published in its current form.

Author Response

Thank you for your comments.

Please find attached, it is responses for reviewer's 3 comments.

Round 2

Reviewer 3 Report

Comments and Suggestions for Authors

I still have some minor comments to share:

I maintain my reservations regarding the characterization of your analysis as a synthesis. It appears to be more of a descriptive analysis rather than a synthesis. Synthesizing research findings involves more than simply assembling quotes; it entails summarizing, evaluating, interpreting, and drawing conclusions from the information. I did not observe much interpretation or evaluation in your analysis. Please reconsider the use of the term "synthesize" or clarify the approach used.

There are a few small errors, such for example in line 121.

Please consider standardizing the terminology used for the sources cited. Currently, you use a variety of terms such as articles, papers, and studies. Using a consistent term throughout would enhance clarity and readability.

Comments on the Quality of English Language

no comments 

Author Response

Thank you for your comments. I have addressed the comments.

"I maintain my reservations regarding the characterization of your analysis as a synthesis. It appears to be more of a descriptive analysis rather than a synthesis. Synthesizing research findings involves more than simply assembling quotes; it entails summarizing, evaluating, interpreting, and drawing conclusions from the information. I did not observe much interpretation or evaluation in your analysis. Please reconsider the use of the term "synthesize" or clarify the approach used".

The term "synthesis" was removed from the scoping review paper.

"There are a few small errors, such for example in line 121". 

It is addressed.

"Please consider standardizing the terminology used for the sources cited. Currently, you use a variety of terms such as articles, papers, and studies. Using a consistent term throughout would enhance clarity and readability".

The terminology "studies" was standardised in the scoping review paper. 

Thanks,